# Acaricidal Biominerals and Mode-of-Action Studies against Adult Blacklegged Ticks, *Ixodes scapularis*

**DOI:** 10.3390/microorganisms11081906

**Published:** 2023-07-27

**Authors:** Grayson L. Cave, Elise A. Richardson, Kaiying Chen, David W. Watson, R. Michael Roe

**Affiliations:** Department of Entomology and Plant Pathology, North Carolina State University, 3230 Ligon Street, Raleigh, NC 27695, USA; glcave@ncsu.edu (G.L.C.); earicha5@ncsu.edu (E.A.R.); kchen23@ncsu.edu (K.C.); wwatson@ncsu.edu (D.W.W.)

**Keywords:** mechanical insecticides, ticks, Imergard, Celite, repellency

## Abstract

Ticks in the USA are the most important arthropod vector of microbes that cause human and animal disease. The blacklegged tick, *Ixodes scapularis*, the focus of this study, is able to transmit the bacteria that causes Lyme disease in humans in the USA. The main approach to tick control is the use of chemical acaricides and repellents, but known and potential tick resistance to these chemicals requires the discovery of new methods of control. Volcanic glass, Imergard, was recently developed to mimic the insecticide mode of action of the minerals from diatoms (diatomaceous earth, DE) for the control of malaria mosquitoes in Africa. However, studies on the use of these minerals for tick control are minimal. In a dipping assay, which was put into DE (Celite), the times of 50 and 90% death of adult female *I. scapularis* were 7.3 and 10.5 h, respectively. Our mimic of DE, Imergard, killed ticks in 6.7 and 11.2 h, respectively. In a choice-mortality assay, ticks moved onto a treated surface of Imergard and died at 11.2 and 15.8 h, respectively. Ticks had greater locomotor activity before death when treated by dipping for both Imergard and Celite versus the no-mineral control. The ticks after making contact with Imergard had the mineral covering most of their body surface shown by scanning electron microscopy with evidence of Imergard inside their respiratory system. Although the assumed mode of action of Imergard and Celite is dehydration, the minerals are not hygroscopic, there was no evidence of cuticle damage, and death occurred in as little as 2 h, suggesting minimal abrasive action of the cuticle. Semi-field and field studies are needed in the future to examine the practical use of Imergard and Celite for tick control, and studies need to examine their effect on tick breathing and respiratory retention of water.

## 1. Introduction

Ticks are a nuisance and an important vector of pathogens for humans and animals, including livestock and companion animals. For example, the blacklegged tick, *Ixodes scapularis* Say (Acari: Ixodidae), in the United States, transmits the bacteria, *Borrelia burgdorferi*, that is the causal agent of Lyme disease. Ticks transmit 95% of all vector-borne diseases in the US, with Lyme disease being responsible for greater than 70% of the cases reported [1,2,3]. *Ixodes scapularis* vectors other pathogens that cause disease, e.g., ehrlichiosis, babesiosis, and anaplasmosis. Chemical pesticides, insecticide-treated clothing, and the application of repellents to skin and clothing are the strategies mostly used to control ticks and to prevent ticks from biting. However, there are concerns about the exposure to pesticides and the growing tick resistance to acaricides [4]. Although resistance has not been found in *I. scapularis* so far, resistance evolution is a possibility [4,5]. Vaccination of animals, such as dogs or rats, which can act as reservoirs of Lyme disease, and cattle, which can be vaccinated against the cattle tick, *Rhipicephalus microplus*, is also used to prevent tick infection [2,6,7,8,9]. With an expanding range for ticks that can possibly vector diseases [10,11] and the possibility that being infected with *B. burgdorferi* could increase the survivability of overwintering ticks [12], alternative methods of control are greatly needed.

Mechanical insecticides are minerals, e.g., diatomaceous earth (DE) produced by diatoms, that have a physical mode of action for killing insects. Mechanical insecticides are active against a variety of insects, including some of medical importance: mosquitoes [13,14], flies [15,16], fleas [17], bedbugs [18], beetles [19,20,21,22], cockroaches [23], and thrips [24], and they could represent a new method for tick control without the concerns associated with chemical pesticides. The hypothesized mode of action of mechanical insecticides is abrasion of the cuticle through a physical action or through the removal of cuticular lipids by absorption, which causes dehydration and death [16,17].

Most recently, the industrial minerals diatomaceous earth (Celite 610) and a newly discovered mechanical insecticide, Imergard^TM^, were shown to be active against larval and nymphal lone star ticks, *Amblyomma americanum* [25,26]. Imergard was originally developed from volcanic glass for adult mosquito control, as a residual wall spray for use in homes in Africa [13,14]. Richardson et al. [27] found Celite and Imergard to be highly efficacious against larval blacklegged ticks. Showler and Harlien [28] found that Celite could be used to prevent larval southern cattle fever ticks, *R. microplus*, from attaching to and feeding on calves. Although these minerals were active against three different species of larval ticks, they were not tested against adult ticks. Also, their mode of action in ticks is not well studied. Richardson et al. [27] provided evidence that the minerals might block the spiracles of nymphal blacklegged ticks. 

The current study examined the efficacy of Celite and Imergard against adult female *I. scapularis* by two exposure methods, dipping and a choice-mortality assay. In addition, mode-of-action studies were conducted by examining the locomotor activity after treatment and the positioning of the minerals on the tick surface, including the openings into the respiratory system (by scanning electron microscopy).

## 2. Materials and Methods

### 2.1. Mechanical Insecticides and Ticks

Unfed adult female blacklegged ticks, *I. scapularis,* were purchased from Oklahoma State University (Oklahoma City, OK, USA). The ticks were approximately 70–90 days post-molt and were used within two weeks of arrival. Until used, the ticks were maintained at 27 ± 1 °C and 70 ± 5% relative humidity with a 14:10 L:D cycle. 

The mechanical insecticides Imergard™ WP and Celite 610 were obtained from the company Imerys Filtration Minerals, Inc. (Roswell, GA, USA) and stored at room temperature in the laboratory in their original packaging until needed. All bioassays were conducted in a Percival I-36NL incubator (Percival Scientific, Inc., Perry, IA, USA) under environmental conditions described later.

### 2.2. Dipping Assay

Ticks were dry-dipped for 1 s in 25 mg of mechanical insecticide, either Celite or Imergard, in the bottom of a 60 mm × 15 mm plastic Petri dish (Fisher Scientific, Hampton, NH, USA) using a round #4 camelhair brush (Craft Smart, Irving, TX, USA). Each tick on the tip of the brush was completely submerged below the surface of the powder. After dipping, the tick was placed into a clean 60 mm × 15 mm plastic Petri dish (one tick per dish), and the dish bottom and top were immediately sealed together along the outer edge with Parafilm (Bemis Company, Neenah, WI, USA). After each dipping, the camelhair brush was cleaned by brushing the surface of a 9 cm-diameter, coarse-grade filter paper (Fischerbrand, Walthem, MA, USA) on a flat benchtop. A new dish containing 25 mg of mechanical insecticide was replaced after treating 10 ticks. For the controls, each undipped tick was transferred with a camelhair brush (that never came in contact with the mechanical insecticides) into a clean Petri dish, and the Petri dish was processed as described for the treatments. The dishes containing treated and control ticks were then incubated at 30 ± 1 °C, 50 ± 5% relative humidity, and during the photophase. These conditions were used so the results would be directly comparable to our previous results with mosquitoes [13] and filth flies [15]. In this study, the ticks were observed every 30 min until all ticks were dead. Death was defined as no tick movement after gently tapping the center of the Petri dish. The treatments and controls were each replicated three times (10 ticks per replicate), and mortality data were analyzed by repetitive sampling (observation of death) every 30 min using a Probit model (described later).

### 2.3. Tick Movement

At 1 h intervals for both the treatments and the controls in the dip bioassays for both Celite and Imergard, the position of the tick on the Petri dish bottom was marked in ink by consecutive numbers on the outside top of the Petri dish. After the ticks were all dead, the distance each tick moved between each mark was measured ±1 mm for each 1 h interval up to the LT_50_ (the time to 50% mortality) and used as an index of tick locomotor activity. Before the LT_50_, any time intervals with no tick movement, the tick was considered dead; this zero movement was not recorded, but the earlier distance moved was included in the analysis.

### 2.4. Choice-Mortality Assay

This assay was conducted in 12 cm × 3.5 cm glass Petri dishes (Fisher Scientific). Scotch blue tape #2090 (3M, St. Paul, MN, USA) was applied to half of the Petri dish bottom (sticky surface to the glass), and 2 g of Imergard was added to the center of the Petri plate bottom and was spread as evenly as possible by sight over the entire surface of the bottom using a metal spatula. The tape with Imergard on top was removed slowly to prevent any transfer of the mineral to the Petri dish glass bottom under the tape. This action produced a plate where half of the bottom contained Imergard, and the other half was Imergard free.

One tick was placed approximately in the center of each dish but on the untreated surface, using a clean camelhair brush (previously described). Then, the top of the Petri dish was applied, and the dish sealed with Parafilm as described earlier. The controls were conducted the same except the Petri dish did not contain Imergard. The dishes containing treated and control ticks were then incubated at 30 ± 1 °C, 50 ± 5% relative humidity and during the photophase except for the last three observations, and the ticks observed every h until all ticks were dead. For the observations during the scotophase, the incubator lights were briefly turned on to score mortality. Death was defined as no tick movement after gently tapping the center of the Petri dish. Five dishes were used per replicate and three replicates were conducted for both the treatments and control. Mortality data were analyzed by repetitive sampling (observations of death) every 60 min using a Probit model (described later).

### 2.5. Scanning Electron Microscopy (SEM)

Ticks from the choice-mortality assay were stored individually at −40 °C until needed for SEM analysis. The SEM analysis was conducted at the Analytical Instrumentation Facility on the North Carolina State University Centennial Campus (Raleigh, NC, USA). The ticks were desiccated in a vacuum for 48 h and then fixed with super glue onto a Hitachi SEM aluminum mount. The samples were subjected to Cressington sputter coating for 60 s with a 70 nm gold–palladium mixture (60 Au/40 Pd). A Hitachi SU3900 (Hitachi, Ltd., Chiyoda City, Tokyo, Japan) variable pressure scanning electron microscope was used to image the tick surface for both treatments and the control.

### 2.6. Statistical Analysis

Mortality data were collated using Excel 2016 (Microsoft Co, Redmond, WA, USA). Probit, ANOVA, and t-tests were performed in JMP Pro 16 (JMP Statistical Discovery LLC, Cary, NC, USA). Abbott’s correction was utilized when necessary, but only one control tick died in the entire study.

## 3. Results

### 3.1. Mortality of Dipped I. scapularis Adults

Figure 1 shows percentage mortality versus time after treatment for adult female *I. scapularis* dipped into Celite (Figure 1A) or Imergard (Figure 1B). For Celite, the first mortality occurred at 3.5 h with 100% mortality obtained after 12 h (Figure 1A). There was no mortality in the controls for the first 10.5 h, and one tick died at 11 h. The data shown in Figure 1A is not Abbot corrected. The probit model (Abbot corrected) for these data are shown in Table 1. The LT_50_ (the time to 50% mortality) and LT_90_ (the time to 90% mortality) for Celite were 7.29 and 10.5 h, respectively, and were statistically significantly different (no overlap of the 95% confidence intervals). 

Imergard killed the ticks in a similar time to Celite. The first mortality occurred at 2 h, and all of the ticks were dead after 17 h, with no mortality in the control (Figure 1B). The LT_50_ and LT_90_ for Imergard were 6.72 and 11.2 h, respectively, and were statistically significantly different (no overlap of the 95% confidence intervals) (Table 1). At the LT_50_ and LT_90_, there was no statistical differences between Celite and Imergard (Table 1). 

### 3.2. Mortality of I. scapularis Adults in the Choice Mortality Assay

The ticks remained on the Petri dish bottom for the entire assay, started dying at 6 h, and were all dead after 17 h (Figure 2, no mortality in the control), which was longer than for Celite and Imergard in the dipping assay (Figure 1A and Figure 1B, respectively). The LT_50_ and LT_90_ for the choice-mortality assay were 11.2 and 15.8 h, respectively, which were statistically significantly different (no overlap of the 95% confidence intervals (Table 1)). The LT_50_ and LT_90_ were also longer than that for Celite and Imergard in the dipping assay (Table 1). In the choice-mortality assay, the LT_50_ and LT_90_ for Imergard were 1.66- and 1.41-fold, respectively, longer than that for Imergard in the dipping assay (Table 1). 

### 3.3. Mineral Exposure by Dipping on I. scapularis Movement

The location of the ticks in the Petri dish after dipping into Celite and Imergard was marked on the dish lid every hour, and the distance between each location was measured for each time interval as an index of the locomotor activity for the treatment compared to the control. The results for Celite for 1–7 h are shown in Figure 3A; data collection was stopped at the LT_50_ (Table 1). Distance moved at 2–4 h for the treatment was higher than the control (ANOVA, F = 7.18, df = 117, *p* > F = 0.0084), but no differences were noted at 5–7 h (ANOVA, F = 0.151, df = 101, *p* > F = 0.699). If you compare the average movement of the treatment versus the control at 2–4 h (Figure 3B), treatment movement was also higher (*t*-test, *t*(115) = 0.0084, *p* = 0.05) than the control, while this was not the case at 5–7 h (*t*-test, *t*(99) = 0.6888, *p* = 0.05). This increased movement over the control was also found for Imergard but at all time points after 1 h through 5 h (Figure 3C, ANOVA, F = 10.4, df = 141, *p* > F = 0.0016; Figure 3D, *t*-test, *t*(137) = 0.0014, *p* = 0.05). Data collection was stopped at the LT_50_ (Table 1). 

### 3.4. Scanning Electron Microscopy of I. scapularis Exposed to Imergard in the Choice Mortality Assay

Figure 4 shows a close up by SEM of Imergard on the tick surface. The mineral is essentially amorphous, with no regular shape. Scanning electron microscopy of Imergard on different parts of the tick body are shown in Figure 5B,D,F compared to untreated ticks (Figure 5A,C,E). The pictures show an even coating on the cuticle with some major clumping present (shown with arrows in Figure 5B,D,F). No evidence of cuticular damage was observed. The spiracular plate and aeropyles (Figure 6A and Figure 6C, respectively) show the normal state of the entrance to the tick tracheal system in the control. In treated ticks, the plate is covered with Imergard (Figure 6B,D), and in some cases Imergard had penetrated into the aeropyle (Figure 6D; far left arrow). 

## 4. Discussion

Two mechanical insecticides, Celite 610 and Imergard WP, were investigated for their acaricidal activity against unfed, adult, female blacklegged ticks, *I. scapularis.* It is well established that minerals produced by some diatoms like that found in Celite (diatomaceous earth, DE) are insecticidal, with far less information about their use to control ticks [29]. A few years ago we developed an alternative to diatomaceous earth, made from volcanic glass, Imergard, as a residual wall spray for mosquito control inside of homes in Africa [13,14]. The advantages of Imergard, it does not contain crystalline silica found in low levels in diatomaceous earth, and volcanic glass is highly abundant. Imergard easily suspends in water, can be applied by spraying using typical equipment used for vector control, and controlled mosquitoes for at least 6 months in phase II field trials in Benin, Africa [14]. 

In a simple dipping assay, we found 50 and 90% of adult female *I. scapularis* ticks were dead in 6.72 and 11.2 h, respectively, when exposed to Imergard; the time to death for the mineral, Celite (diatomaceous earth), was 7.29 and 10.5 h, respectively, not significantly statistically different from Imergard (Table 1). Showler et al. [25] and Showler and Harlien [26] found Celite and Imergard were able to control larval and nymphal lone star ticks, *A. americanum.* Richardson et al. [27] found Celite and Imergard were highly efficacious against nymphal blacklegged ticks. Showler and Harlien [28] found that Celite could be used to prevent larval southern cattle fever ticks, *R. microplus*, from attaching and feeding on calves. Although these minerals were active against three different species of ticks in the above published studies, they were not tested against adult ticks. The current study is the first evidence that Celite and Imergard could control unfed, female adult blacklegged ticks.

Richardson et al. [27] used the same dipping assay, tick age after molting (50–70 days) and environmental conditions post treatment with Imergard and Celite on unfed nymphal, blacklegged ticks (mixed sexes) as that used in the current study for adult ticks. Comparing the different stages, adult ticks appear to be less susceptible than nymphal ticks, both about the same age after molting. For example, the LT_50_ and LT_90_ were 5.4- and 4.9-fold, respectively, longer for adult females than nymphs for Celite and 5.7- and 6.0-fold, respectively, longer than nymphal ticks for Imergard. Richardson et al. [27] also found Imergard was slightly more effective (had a shorter LT_50_ and LT_90_) than Celite in nymphal *I. scapularis* but this was not the case for adult females of the same species (Table 1). The mode of action of mechanical insecticides like Celite is thought to be through physically damaging the cuticle, resulting in water loss [16,17], that upsets osmotic balance and finally causes death. This could occur by the abrasive action of the mineral disrupting the outer water proofing wax layer of the cuticle as body parts rub against each other and/or by rubbing against solid substrates as the insect moves; and/or the mineral absorbing the wax layer of the cuticle. Deguenon et al. [13,14] suggested the mechanism of abrasion was not likely in mosquitoes for Imergard. When they landed on an Imergard treated surface, the mineral transferred mostly to the lower legs, presumably by electrostatic charge differences, the insect quickly became quiescent, and then the mosquito died demonstrating symptoms similar to pyrethroid poisoning. 

Chen et al. [15] found that for three species of filth flies exposed in a modified WHO cone test to a surface treated with Imergard, the entire body was covered with minerals, completely different from that for mosquitoes [13]. The flies also were more active after exposure unlike mosquitoes. More research is needed to better quantify this increased fly activity. Adult *I. scapularis* females were similar to flies in two ways. The choice-mortality assay showed their entire bodies were covered with mineral (Figure 5), and after the dipping-mortality assay, the index for adult movement increased compared to controls (Figure 3). This increased movement could be the result of the mineral being an irritation to the ticks, and movement is a method to reduce this irritation. Richardson et al. [27] also found blacklegged nymphs dipped into Celite had all of their body surfaces covered with the mineral; however, activity levels after dipping were not measured. In the case of nymphs, they started dying in 10–20 min, suggesting abrasion was not a factor in this short time frame and where tick overall movement appears by observation to be significantly less than filth flies. Increased activity levels in ticks and filth flies after treatments do suggest they are able to detect the mineral on their body surface by some method that is not currently known. In the case of flies, the increase in activity occurred in seconds after mineral application, while for adult, female *I. scapularis*, increased movement was detected after 1 h (Figure 3). 

In the Chen et al. [15] filth fly studies, Imergard killed the smaller blow flies and house flies faster than the larger flesh flies. The same relationship between weight and time to death was found if you compare the time to death for smaller blacklegged nymphal ticks [27] to that for larger adult female *I. scapularis*; the fold differences for Imergard and Celite were mentioned earlier in this discussion. Volume increases by the cube, surface area by the square as the size of an insect or tick increases (where the body shape is similar). The surface/volume ratio of the adult female tick is smaller relatively to that for nymphs. The hypothesis would be that a topically active pesticide that causes dehydration would have a greater effect for nymphs as our research showed. Dawson [30] also reported that diatomaceous earth was more active for a smaller body size. 

Richardson et al. [27] found that the time to death for older unfed *I. scapularis* nymphs was longer than younger unfed nymphs and hypothesized that this resulted from more mature cuticle with greater protection from water loss by an enhanced wax layer as one possibility. This result with older nymphs was unexpected since these ticks do not consume water before blood feeding. The older ticks would be expected to be more dehydrated and more susceptible to a mechanical insecticide that disrupts the water barrier of the cuticle. Although we have no data to support this hypothesis, there may be some evidence to support this idea between unfed nymphal and adult ticks. Yoder et al. [31] found 3.6-fold greater cuticular lipid in 30 d post-molt, adult versus nymphal lone star ticks, *A. americanum*. If this increase in the wax layer occurs between nymphal and adult *I. scapularis* as well, this might explain the longer time to death after dipping into Celite and Imergard for the blacklegged tick adult. There are also fundamental differences in adult female cuticle compared to nymphs, where the former must be prepared to increase their body size several fold when feeding to repletion. 

Richardson et al. [27], in dipping-mortality assays, found that, in some cases, the blacklegged nymphal tick spiracular sieve plate was heavily blocked with Celite. In the current study with adult female *I. scapularis* in the choice-mortality assay, the plate was also covered with Imergard at high levels in some cases (Figure 6B). Since the sieve plate was visible in some treatments, we were able to view the aeropyles and found Imergard inside of the small openings into the respiratory system (Figure 6D). This could have also been the case for Celite in the studies by Richardson et al. [27], where it was hypothesized that the rapid death of ticks in 20 min or less might partly be a result of blockage of the respiratory system. 

Finally, the choice-mortality assay for *I. scapularis* female adults is encouraging, suggesting Imergard could be used as a residual treatment for tick control not requiring direct contact with the tick at the time of application. If Imergard had any repellency, it was not enough to prevent them from walking onto a treated surface in our choice-mortality assay and obtaining a lethal dose. These results are consistent with those of Deguenon et al. [13] that found Imergard treated surfaces were not repellent to mosquitoes or Chen et al. [15] which found Imergard was not repellent to three species of filth flies to the point of preventing the acquisition of a lethal dose. Semi-field and field studies are needed in the future to determine the practical use of Imergard and Celite for the control of the blacklegged tick and other tick species. Showler and Harlien [28] found that Celite could be used to prevent larval southern cattle fever ticks, *R. microplus*, from attaching and feeding on calves, providing additional evidence of the practical use of these mechanical insecticides for tick control.

## 5. Conclusions

In conclusion, both Celite 610 and Imergard WP in dipping assays were lethal to adult, unfed, female *I. scapularis* ticks in the laboratory. The mechanical insecticide Imergard in choice-mortality assays was also lethal, demonstrating that the mineral repellency, if any, was not enough to prevent them from walking onto a treated surface and receiving a lethal dose. Celite and Imergard treatments increased tick locomotor activity. Ticks also, after walking onto a treated surface of Imergard, had their entire bodies covered with the mineral with evidence of mineral intrusion into the sieve-plate aeropyles, the entrance to their respiratory system. Further semi-field and field studies are needed to determine the practical use of these minerals in the control of the Lyme disease tick and other species.

## Figures and Tables

**Figure 1 microorganisms-11-01906-f001:**
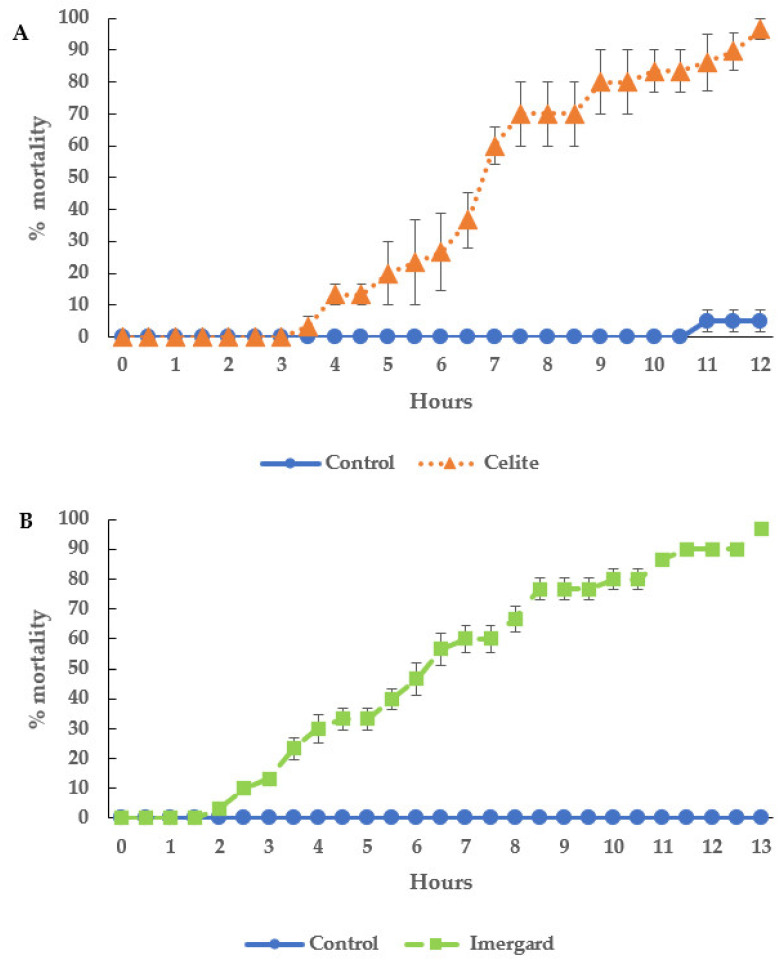
Mortality of unfed adult female *Ixodes scapularis* dipped in (**A**) Celite and (**B**) Imergard and then incubated in the photophase at 30 ± 1 °C and 50 ± 5% relative humidity. Error bars are ±1 SEM. In some cases, the error bars were too small to be visible.

**Figure 2 microorganisms-11-01906-f002:**
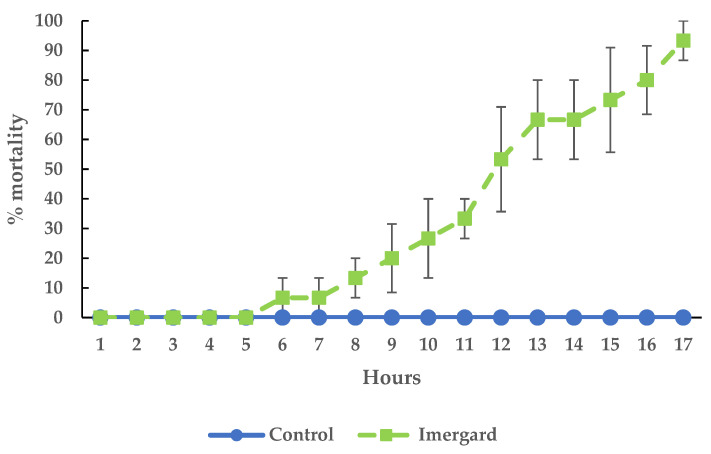
Mortality of *Ixodes scapularis* in the choice-mortality assay. The assay was conducted at 30 ± 1 °C and 50 ± 5% relative humidity during the photophase except for the last 3 h. Error bars are ±1 SEM. In some cases, the error bars were too small to be visible.

**Figure 3 microorganisms-11-01906-f003:**
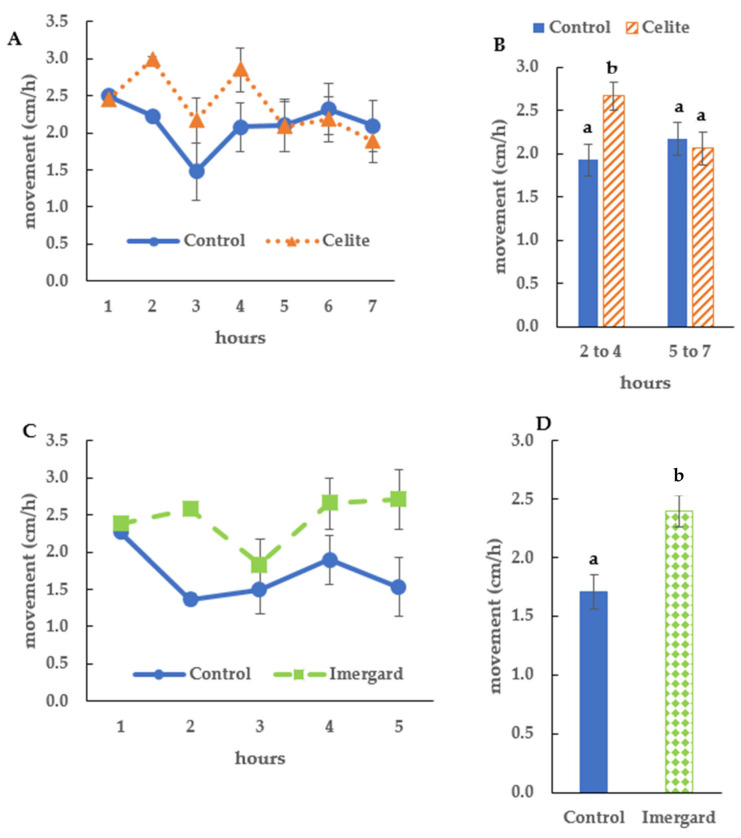
(**A**) Movement of Celite dipped versus control ticks, (**B**) the average distance traveled of Celite dipped versus control ticks between two periods of movement from 2–4 and 5–7 h, (**C**) movement of Imergard dipped versus control ticks from 2–5 h, and (**D**) the average distance traveled of Imergard dipped versus control ticks during 2–5 h. In (**B**,**D**), different lower case letters indicated a statistically significant difference as determined by a *t*-test (see results for details). Error bars are ±1 SEM. In some cases, the error bars were too small to be visible.

**Figure 4 microorganisms-11-01906-f004:**
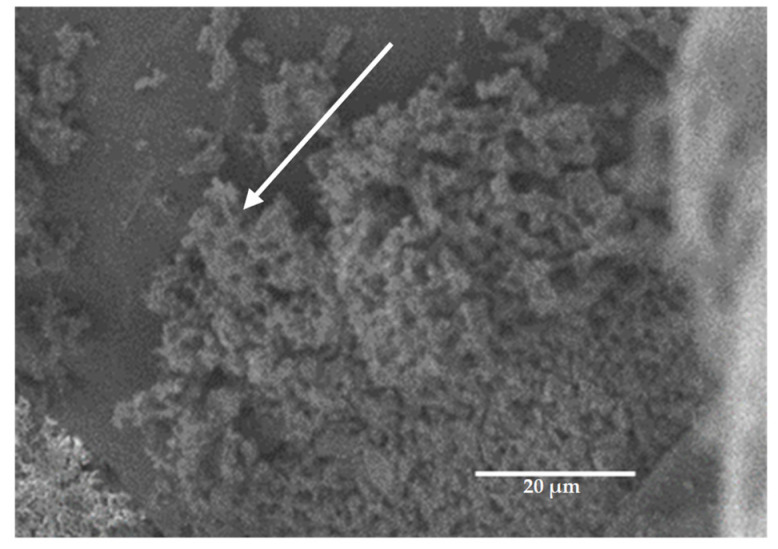
Close view by scanning electrosn microscopy of Imergard (arrow) on the surface of *Ixodes scapularis*.

**Figure 5 microorganisms-11-01906-f005:**
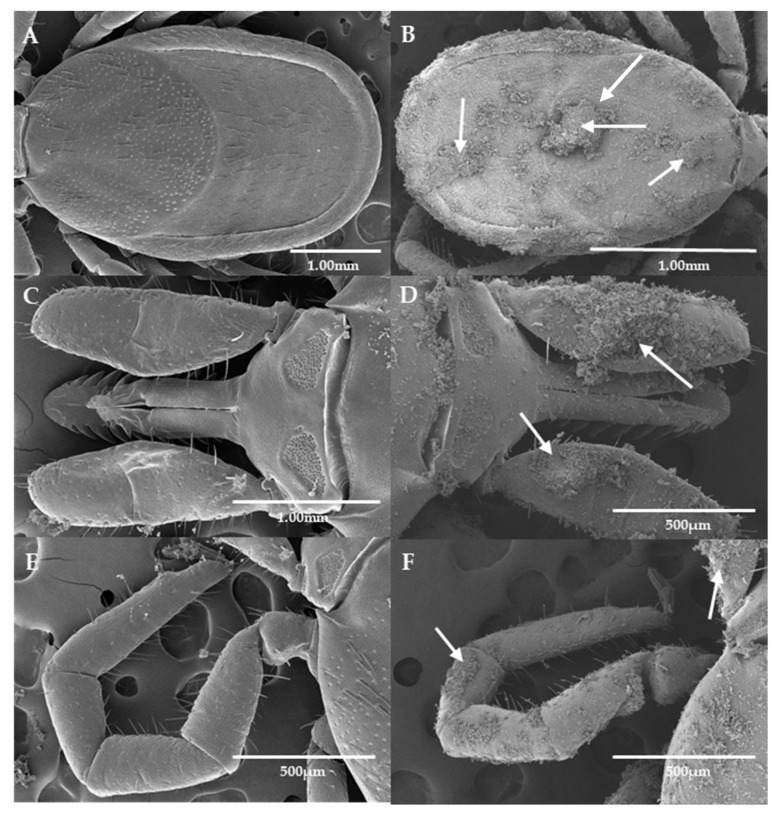
Scanning electron microscopy of control (**left**) and treated (**right**) *Ixodes scapularis* in the Imergard choice-mortality assay. (**A**,**B**) full body, (**C**,**D**) capitulum, and (**E**,**F**) leg. Arrows show clumping of Imergard. The reason for clumping is unknown.

**Figure 6 microorganisms-11-01906-f006:**
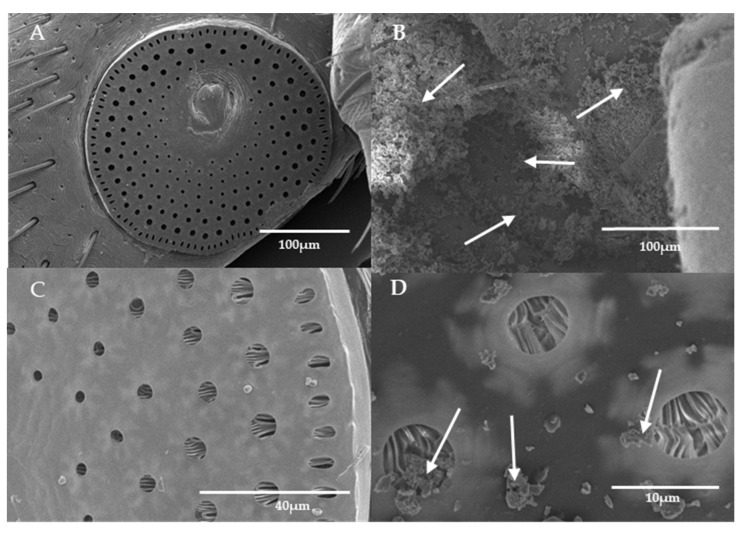
Scanning electron microscopy of the (**A**) control spiracular plate (round structure with holes), (**B**) treated spiracular plate, (**C**) control aeropyles (holes in the spiracular plate), and (**D**) treated aeropyles of *Ixodes scapularis* in the Imergard choice-mortality assay. Arrows in (**B**) show Imergard blockage of the openings in the spiracular plate. Arrows in (**D**) show Imergard inside of an aeropyle (far left arrow) or otherwise on the surface of the spiracular plate.

**Table 1 microorganisms-11-01906-t001:** Probit models for *Ixodes scapularis* in the dipping and choice-mortality assays †.

Mechanical Insecticide	N	Slope (SEM)	LT_50_ ^†^ (95% CL)	LT_90_ ^†^ (95% CL)	Χ^2^
Celite 610 dip	30	0.399 (0.0246)	7.29 A ^‡^ (6.98–7.60)	10.5 A ^‡^ (10.0–11.1)	17.56
Imergard dip	30	0.358 (0.0310)	6.72 A (6.35–7.09)	11.2 A (10.6–11.9)	8.90
Imergard choice-mortality	15	0.280 (0.0321)	11.2 B (10.5–12.0)	15.8 B (14.6–17.5)	2.32

† LT_50_ and LT_90_ is the time in h to 50 and 90% mortality, respectively (CL = confidence limit; Χ^2^ = chi-square). ‡ Values in the same column with overlapping 95% CLs are considered not statistically significant and are designated by the same letter.

## Data Availability

Data available upon justifiable request by the corresponding author.

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
