# Peer review of "Acaricidal Biominerals and Mode-of-Action Studies against Adult Blacklegged Ticks, Ixodes scapularis"

_microorganisms, 2023, doi:10.3390/microorganisms11081906_

Round 1
Reviewer 1 Report
The article “Acaricidal Biominerals and Mode of Action Studies against Adult Blacklegged Ticks, Ixodes scapularis” provide a very interesting study on the effects of diatomaceous earth analogues against Ixodes scapularis. Overall, the manuscript is well written, although I suggest major changes to the introduction and material and methods sections. Also, I wouldn’t call these experiments “mode of action studies”, in my opinion more molecular analysis are needed to demonstrate that; the authors are showing its effects but can only suggest a mode of action. In the introduction, I would add background on biominerals, at least what they are and since when they have been used as acaricides. In materials and methods, there are some points to clarify and, although I think the discussion is good and interesting, I would consider reviewing the concepts of repellency and acaricidal activity through the manuscript. Repellency was not the focus of the study, but it is mentioned later in the discussion. It is confusing. See my comments for more details.

Reviewer 2 Report
The manuscript, “Acaricidal Biominerals and Mode of Action Studies Against Adult Blacklegged Ticks, Ixodes scapularis,” describes studies designed to test mineral dusts (i.e. Celite or Imergard) for potential acaricidal efficacy against unfed adult female blacklegged ticks (Ixodes scapularis). This is a significant study because: i) ticks transmit a variety of pathogens to humans and animals, ii) have shown increasing development of resistance to chemical acaricides, iii) their potential habitat range is likely increasing due to climate change, and iv) there is evidence that tick infection by pathogens may increase their ability to survive overwintering conditions. Taken together, these factors significantly increase the risk of tick-borne disease transmission to both humans and other animals. The mineral dusts utilized in the present study are believed to act mechanically by disruption of the cuticular barrier to water loss or by blockage (and potential damage) to the invertebrate respiratory system, rather than by toxic effects on metabolic activity. The current report extends previous studies on larval and nymphal ticks, demonstrating acaricidal activity to adult ticks, and providing additional information on likely mode of acaricidal action. Overall, the manuscript is well written and organized, presenting the findings in a clearly understandable manner. There are listed below, however, a few corrections and minor suggestions to improve the present manuscript.
Figure 1B does not show that all ticks are dead at 8.5 hrs as described in the text (lines 157-158). This should be corrected.
Section 3.3 describes how tick movement was recorded as the distance between marked tick positions moved during each time interval. There was not continuous monitoring of tick movement, therefore the recorded measurements reflect only a minimum value of actual tick movement. This fact should be noted within the text and discussion relating to measurement and comparison of tick movement between the experimental and control conditions.
Figures 5-6: Although the authors note clumping of Imergard (arrows), they fail to note any potential explanation of how or why this observed clumping may be significant (could indicate areas sticking to hemolymph or moisture escaping due to cuticular damage).
Discussion: Although the authors have noted one potential mode of action by loss of moisture, they did not test this hypothesis by incubation under differing relative humidities (low/medium/high), which might provide evidence for this mode of action by revealing differences in survival under different relative humidities. Also (lines 290-292), the observation of increased movement following exposure to the mineral dusts suggests that the treatment was irritating to the ticks. As previousy noted, the method of measuring movement (Section 3.3) provided only a minimum measure of actual tick movement.
Conclusion: Good, except that an additional study of the effects of the mineral dusts under different conditions of relative humidity could provide additional insight into the potential mode(s) of action of the mineral dusts. An additional future study might be on-animal application of the mineral dusts (e.g., dust bags or other application) and potential hazards and safety to animals (external application, but also inhalation safety).
Round 2
Reviewer 1 Report
The authors addressed the comments and clarified the suggested points . Nothing else to add.